# On the Generalisation of Koopman Representations for Chaotic System Control

Kyriakos Hjikakou[1], Juan Diego Cardenas Cartagena[1], and Matthia Sabatelli[1]

[1]University of Groningen, Department of Artificial Intelligence, Groningen, Netherlands

## Abstract

This paper investigates the task generalisability of Koopman-based representations for chaotic dynamical systems, focusing on their transferability across prediction and control tasks. Using the Lorenz system as a testbed, we propose a three-stage methodology: learning Koopman embeddings through autoencoding, pre-training a transformer on next-state prediction, and fine-tuning for safety-critical control. Our results show that Koopman embeddings outperform both standard and physics-informed PCA baselines, achieving accurate and data-efficient performance. Notably, fixing the pre-trained transformer weights during fine-tuning leads to no performance degradation, indicating that the learned representations capture reusable dynamical structure rather than task-specific patterns. These findings support the use of Koopman embeddings as a foundation for multi-task learning in physics-informed machine learning. A project page is available at https://kikisprdx.github.io/.

## 1 Introduction

Neural networks (NNs) have demonstrated effectiveness in modelling chaotic dynamics since the work of Navone and Ceccatto [1], who showed that NNs can be as effective as regressive and statistical methods in certain simplified cases, such as simulating a 1D Lorenz system. A trend in modern chaos modelling has been the integration of physics-informed methods with deep learning architectures. Of particular importance is Koopman operator theory [2], which provides a mathematical framework for transforming non-linear dynamical systems into linear representations [3]. This linearisation enables the development of more stable and interpretable models of chaotic dynamics, with recent work demonstrating that transformer architectures, combined with Koopman embeddings demonstrate competitive performance in autoregressive prediction across multiple chaotic systems [4] and can even incorporate control inputs directly [5]. The transformer's ability to capture long-range temporal dependencies makes it particularly well-suited for modelling such dynamics [6], while Koopman embeddings provide theoretical grounding and improved generalisation across

systems. While these developments have led to substantial improvements in next-state prediction, the extent to which the resulting representations support downstream tasks such as control remains an open question. In Natural Language Processing (NLP), a common paradigm involves pre-training models on next-token prediction tasks, such as estimating $p(x_{t+1}|x_{\leq t})$, followed by fine-tuning on downstream objectives [7, 8]. This strategy has proven highly effective in yielding generalisable and reusable representations across diverse tasks and domains [7]. Inspired by this success, recent work in robotics has also begun to adopt similar approaches: transformers pre-trained on data prediction tasks are repurposed for control and manipulation [9]. However, the potential of such transfer learning strategies has, to the best of our knowledge, not yet been explored in the context of chaotic dynamical systems. Notably, there exists a structural parallel between sequential prediction in NLP and state forecasting in dynamical systems, where the goal is to estimate $p(s_{t+1}|s_{\leq t})$ from past observations [4]. Here, $s_t$ denotes the system's state at time $t$, analogous to a token $x_t$ in a sequence. This analogy suggests that pre-training on next-state prediction may serve as a powerful pretext task for learning representations that transfer to downstream objectives in chaotic systems. To evaluate this possibility, we investigate whether Koopman embeddings learned from a self-supervised prediction task can be reused for downstream control. Effective transfer would suggest that these embeddings capture genuine physical structure rather than task-specific artefacts, while poor transfer would imply limited generalisability across tasks. In addition to assessing the effectiveness of transfer, this question is also practically relevant: if such embeddings are reusable, expensive physics-informed representations could be amortised across multiple tasks, leading to more efficient model development pipelines alongside shorter training times. We assess this hypothesis through empirical evaluation on the popular Lorenz system, comparing the performance of Koopman embeddings on a safety-critical control task[1] against two types of Principal Component Analysis (PCA) baselines: one purely data-driven and another incorporating system-specific physical

---

[1]We use 'control' terminology following literature convention, though our focus is on safety function estimation that would serve as a basis for control implementation [10–12].

Proceedings of the 7th Northern Lights Deep Learning Conference (NLDL), PMLR 307, 2026.

priors. Furthermore, we contrast fixed and fine-tuned transformer configurations to isolate the contribution of the learned embedding from that of the downstream optimisation process.

**Contributions:** The main contributions of this work are threefold. First, it introduces a novel perspective on transfer learning in chaotic dynamical systems by drawing a conceptual parallel with natural language processing, where pre-training on next-token prediction has proven highly effective for enabling downstream generalisation. This analogy motivates the use of next-state prediction as a pretext task for learning reusable representations in physical domains. Second, it demonstrates that Koopman embeddings trained on next-state prediction generalise effectively to downstream control, supporting the hypothesis that such representations encode transferable physical structure. Third, it presents a structured evaluation framework based on PCA and transformer ablations, isolating the contribution of physics-informed structure to transfer performance and emphasising the importance of embedding design in cross-task generalisation.

## 2 Preliminaries

To investigate the transferability of learned representations in chaotic systems, we begin by reviewing the key modelling challenges and mathematical tools underlying our study. We first describe the Lorenz system, a classical benchmark for studying chaos, followed by a discussion of Koopman operator theory, which provides a principled approach to linearising non-linear dynamics. Finally, we define the two core tasks used to assess the generalisability of learned representations: next-state prediction and safety function approximation.

### 2.1 Modelling Chaotic Systems

Chaotic systems pose fundamental challenges for machine learning due to their intrinsic non-linearity and extreme sensitivity to initial conditions. In such systems, small perturbations in state can lead to exponential divergence in future trajectories, making accurate long-term prediction difficult and rendering learned models highly unstable [13]. This behaviour complicates the development of generalisable and reusable representations, as even minor errors in modelling can lead to drastic qualitative changes in system behaviour. The Lorenz system is a canonical example of deterministic chaos and has been widely studied as a benchmark for evaluating machine learning approaches in non-linear dynamics [1, 4, 12, 14]. Its persistent chaotic behaviour across a wide range of initial conditions makes it a suitable testbed for evaluating the transferability of learned

representations. In particular, the consistency of its dynamic complexity [13], ensures that any observed transfer effects can be attributed to properties of the learned embeddings rather than variability in the underlying system dynamics. The Lorenz system is defined by a set of ordinary differential equations where its dynamics are characterised by a strange attractor, a bounded set in the system's phase space to which chaotic trajectories converge despite their non-periodic and highly sensitive nature [15]. The specific ordinary differential equations, alongside implementation details and adopted numerical integration settings, are provided in Appendix A.1.

### 2.2 The Koopman Operator: Linearising Non-linearity

Given the non-linear and chaotic nature of the Lorenz system, analysing or predicting its behaviour directly in the original state space can be challenging. This motivates the use of alternative representations that simplify the system's dynamics. A particularly powerful approach is offered by Koopman operator theory, which provides a linear perspective on non-linear dynamical systems. The central idea behind Koopman theory is that although one might observe the dynamics of physical systems in their natural coordinates, where they typically exhibit non-linear behaviour, there exists a transformation into a space of observables in which the same dynamics evolve linearly. This transformation facilitates analysis and prediction using linear methods, even for inherently non-linear systems. More formally, for a discrete-time dynamical system $s_{t+1} = F(s_t)$, the Koopman operator $\mathcal{K}$ acts on observable functions $g(s)$ (rather than on the states directly) such that:

$$\mathcal{K}g(s) = g(F(s)), \tag{1}$$

implying that the evolution of observables follows a linear rule:

$$g(s_{t+1}) = \mathcal{K}g(s_t). \tag{2}$$

This perspective enables a linear treatment of otherwise complex systems. Notably, Geneva and Zabaras [14] have demonstrated that learning such transformations from data can yield stable and interpretable representations for modelling physical systems. In this work, we leverage Koopman theory to study the generalisation properties of learned dynamical representations, focusing in particular on whether this linearisation facilitates robust forecasting across chaotic trajectories.

### 2.3 Task Formulation

We consider two distinct regression tasks derived from the Lorenz system, both operating on the same input states but differing in the temporal scope of

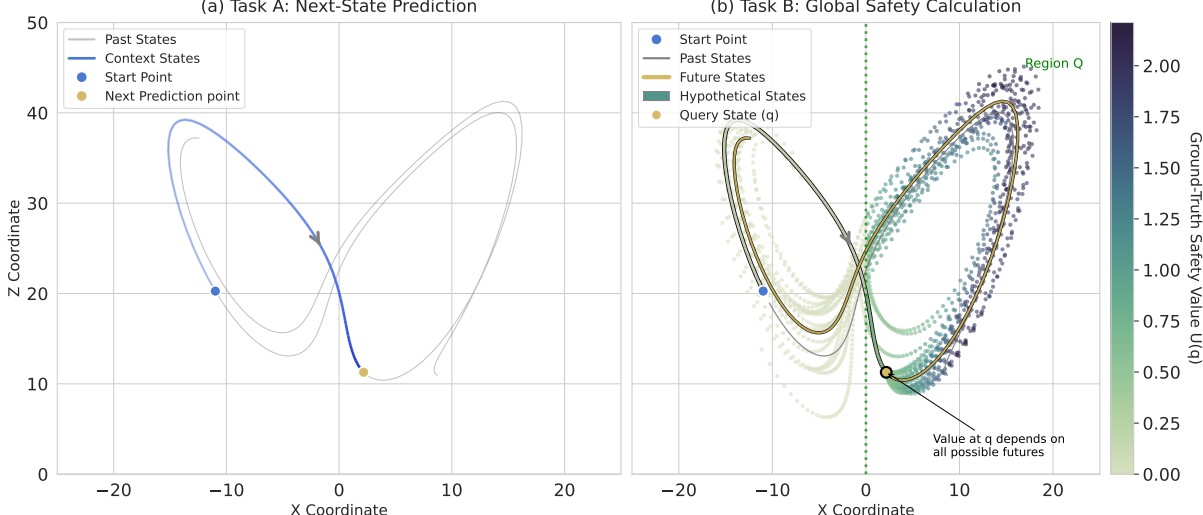

**Figure 1.** Conceptual visualisation illustrating the local, short-horizon nature of next-state prediction (Task A) with the global, long-horizon calculation of the safety function (Task B). Panel (a) illustrates Task A, where a model predicts the single "Next Prediction State" (yellow) using only a finite history of preceding "Context States" (blue). In contrast, panel (b) shows that determining the safety value $U(q)$ for the same "Query State" requires a global understanding of system dynamics. This is represented by the multiple "Hypothetical States" (the set of disturbed positions reachable from the current query state under possible disturbances), where their colour maps to the ground-truth safety value, indicating the risk as they evolve through safety region $Q$.

prediction and the type of system understanding they require.

**Task A: Next-state prediction** is a local forecasting task that models the immediate evolution of the system according to $s_{t+1} = F(s_t)$. Success in this task depends primarily on capturing short-term dependencies and local dynamics [4, 13]. It serves as a natural self-supervised objective to induce predictive structure in learned representations of chaotic systems.

**Task B: Safety function prediction**, in contrast, is inherently global. The goal is to approximate a function $U(q)$ that quantifies the minimum control effort needed to ensure that a trajectory originating at state $q$ remains within a predefined safe region $Q$ indefinitely. In this work, we defined region $Q$ by the bounds $x \in [0, 50], y \in [-50, 50], z \in [-50, 50]$. We chose this region to encompass the likely location of the Lorenz strange attractor of the right wing. Moreover, this task, which is based on the framework of Valle et al. [12], requires the model to implicitly account for the system's behaviour over an unbounded time horizon, including its sensitivity to disturbances. Formally, the safety function corresponds to the converged solution $U_\infty$ of the recursive relation, which is independent of initialisation:

$$U_{k+1}(q_i) \leftarrow \max_{\xi_s} \min_{q_j \in Q} \left( \max \left( ||F(q_i, \xi_s) - q_j||, U_k(q_j) \right) \right)$$

(3)

where $q_i$ and $q_j$ denote discrete state samples, $\xi_s$ represents bounded noise and $F(q_i, \xi_s)$ represents the system dynamics under noise. Although this function is computed through a finite numerical

approximation over a discretised state space (see Appendix A.4), it remains a substantially more challenging and global task than next-state prediction. While the latter requires understanding local transitions, the safety function necessitates a model that encodes the structural knowledge of long-term trajectories and the system's attractor geometry. The contrast between these local and global prediction tasks forms the basis of our transfer learning evaluation, as illustrated in Figure 1.

## 3  Methodology

This section details the experimental framework used to evaluate whether representations learned through next-state prediction in Koopman space can transfer to downstream safety value prediction tasks. We begin by describing the Koopman embedding model architecture, which provides the structural inductive biases for encoding the Lorenz system dynamics. Subsequently, we outline our experimental design, covering data generation and preprocessing, training protocols, baseline model comparisons, and evaluation methods.

### 3.1  Koopman Embedding Implementation

Building upon the theoretical framework introduced in Section 2.2, we start by implementing a neural network-based finite-dimensional approximation of the Koopman operator, aimed at linearising the

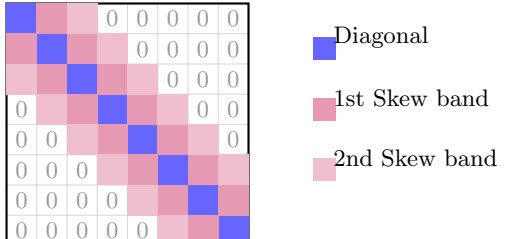

**Figure 2.** Example double-banded skew-symmetric Koopman operator structure as defined by Equation 5 (shown here as $8 \times 8$ - actual implementation uses $32 \times 32$).

chaotic dynamics of the Lorenz system in a learned latent space. The model architecture consists of three components: an encoder $\phi_e : \mathbb{R}^3 \rightarrow \mathbb{R}^{32}$ that maps Lorenz states to a 32-dimensional embedding space, a decoder $\phi_d : \mathbb{R}^{32} \rightarrow \mathbb{R}^3$ that reconstructs the original states from this latent representation, and a Koopman operator $\mathbf{K} \in \mathbb{R}^{32 \times 32}$ that governs linear evolution in the embedding space. See Appendix A.2 for implementation specifications. The learning objective enforces that

$$\phi_e(s_{t+1}) \approx \mathbf{K}\phi_e(s_t), \quad (4)$$

thereby promoting linear predictability of future states in latent space. Inspired by Geneva and Zabaras [4], we do not learn the operator $\mathbf{K}$ as a fully dense matrix. Instead, we impose a physically motivated structure that decomposes $\mathbf{K}$ into the sum of two components:

$$\mathbf{K} = \mathbf{D} + \mathbf{S}_{\text{band}}, \quad (5)$$

where $\mathbf{D} \in \mathbb{R}^{32 \times 32}$ is a diagonal matrix representing element-wise growth and decay rates, and $\mathbf{S}_{\text{band}} \in \mathbb{R}^{32 \times 32}$ is a banded skew-symmetric matrix capturing local rotational dynamics in latent space.

This structured decomposition introduces a strong inductive bias that reflects underlying physical phenomena. The diagonal component $\mathbf{D}$ models dissipative dynamics, such as energy decay, whereas the skew-symmetric bands of $\mathbf{S}_{\text{band}}$ capture conservative oscillatory behaviour reminiscent of rotational or periodic dynamics [16–18]. This separation facilitates the modelling of complex non-linear systems such as the Lorenz attractor, enabling the encoder to discover compact representations of both energy-preserving and dissipative processes [3]. Figure 2 illustrates the structure of the Koopman operator used in our implementation, with colour-coded entries denoting the diagonal and the first two skew-symmetric bands.

### 3.2 Dataset Generation and Splitting

We generated datasets from the Lorenz system using the RK45 integration method ($dt = 0.01$), yielding

2,048 training trajectories (256 steps each), 64 validation trajectories (1,024 steps each), and 256 test trajectories (1,024 steps each). For sequence length $N$, next-step prediction uses time steps 0 to $N-1$ as input to predict steps 1 to $N$. Ground-truth safety values are computed on a 27,000-point discretised grid using Equation 3 [10–12]. These safety values are generated from the same Lorenz system dataset, matching trajectory-wise. However, only 252 test trajectories are included, as four trajectories were entirely outside our predefined safety region $Q$.

### 3.3 Training Protocol

The training process consists of three sequential stages, each corresponding to a distinct model component and task: Koopman autoencoder training, transformer pre-training, and fine-tuning for safety function prediction. These stages are designed to progressively encode the dynamics of the Lorenz attractor into a transferable embedding space, model its evolution over time, and finally adapt this knowledge to a downstream control-relevant prediction task [19–21]. The various shared components and training objectives across the stages, are illustrated in Figure 3. Furthermore, hyperparameter optimisation, training and infrastructure considerations are provided in Appendices A.5, A.6 and A.7, respectively.

In the first stage, the Koopman autoencoder's objective is to learn a latent representation in which the chaotic system dynamics evolve approximately linearly over time. This embedding forms the foundation for downstream predictive modelling. Training uses 64-step sequences with a 16-step stride, yielding 75% overlap across 2,048 temporally shuffled sequences. The encoder $\phi_e$ maps 3D Lorenz states to a 32-dimensional Koopman embedding space, while the decoder $\phi_d$ reconstructs the original states. The neural implementation of Koopman embeddings requires a multi-component loss function to ensure both accurate state reconstruction and proper linear dynamics learning. Building on the work of Geneva and Zabaras [4], we formulate a composite objective function that balances three critical components: reconstruction fidelity, dynamics prediction accuracy, and operator regularisation:

$$\mathcal{L} = \sum_{i=1}^{D} \sum_{j=0}^{T-1} \Big[ \lambda_0 \text{MSE}(s_j^i, \phi_d(\phi_e(s_j^i)))$$
$$+ \lambda_1 \text{MSE}(s_{j+1}^i, \phi_d(\mathbf{K}\phi_e(s_j^i)))$$
$$+ \lambda_2 ||\mathbf{K}||^2 \Big] \quad (6)$$

where $D$ represents the number of trajectories, $T$ the timesteps per trajectory, $i$ indexes individual trajectories, $j$ indexes timesteps within each trajectory,

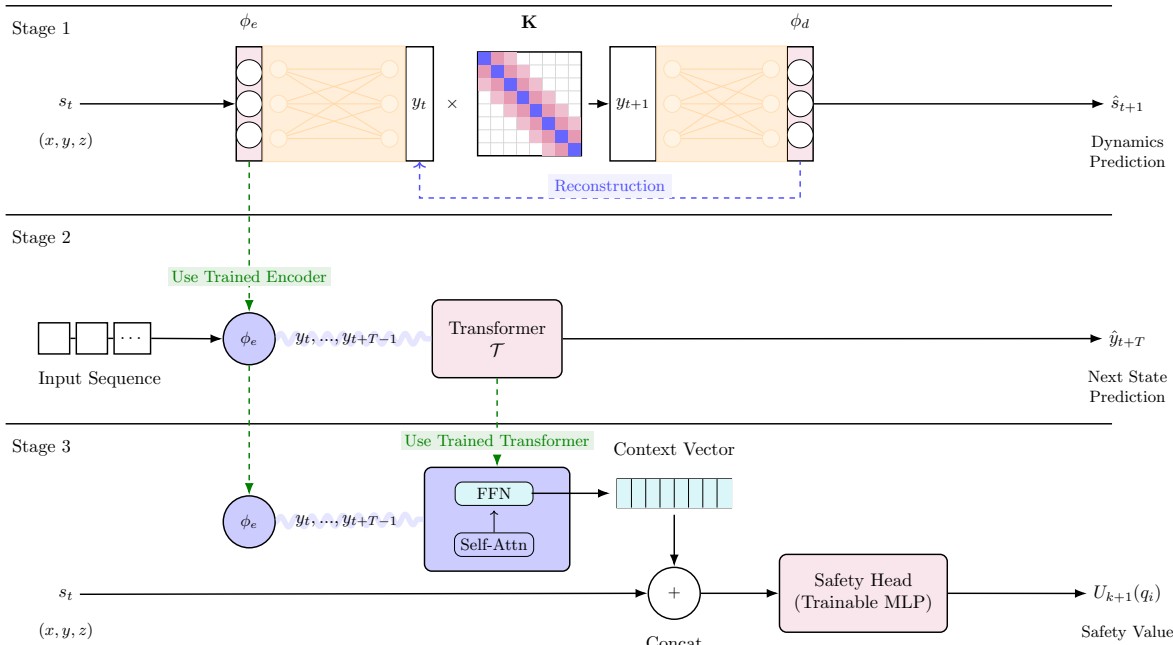

**Figure 3.** An illustration of the three-stage training methodology, which is detailed in Section 3.3. **Stage 1 (Representation Learning):** A Koopman autoencoder is trained to learn a latent representation where the system's dynamics evolve linearly. An input state $s_t$ is mapped by an encoder $\phi_e$ to a latent vector $y_t$. This latent state is then propagated forward by the Koopman operator matrix $\mathbf{K}$, and the decoder $\phi_d$ produces the next-state prediction $\hat{s}_{t+1}$. The model is trained to minimise both dynamics prediction and state reconstruction error. **Stage 2 (Transformer Pre-training):** The encoder $\phi_e$ is frozen and used to convert a sequence of input states into a sequence of latent embeddings. A transformer model $\mathcal{T}$ is then pre-trained on this latent sequence to perform autoregressive next-state prediction (Task A). **Stage 3 (Transformer Fine-tuning):** The pre-trained transformer's weights are frozen. A new NN prediction head is attached, which takes the transformer's final hidden state and a query state $q$ as a concatenated input. This head is then fine-tuned to predict the safety value (Task B).

and $\lambda_0, \lambda_1, \lambda_2$ are hyperparameters weighting the loss components. Term one ($\lambda_0$) enforces reconstruction fidelity by ensuring the encoder-decoder pipeline accurately reconstructs original states. Term two ($\lambda_1$) constitutes the dynamics loss, enforcing proper Koopman operator evolution $\phi_e(s_{j+1}) \approx \mathbf{K}\phi_e(s_j)$. Term three ($\lambda_2$) provides L2 regularisation on the operator matrix to prevent overfitting by encouraging simpler linear dynamics [4]. Finally, the individual loss components use mean squared error (MSE) defined as:

$$\text{MSE}(y, \hat{y}) = \frac{1}{n} \sum_{i=1}^{n} ||y_i - \hat{y}_i||^2, \qquad (7)$$

where, $n$ denotes the number of evaluation samples, $\hat{y}_i$ the predicted value, $y_i$ the corresponding ground-truth value. The full hyperparameters for training the autoencoder are listed in Appendix A.6.

In the second stage, a decoder-only transformer is pre-trained to model Koopman dynamics in embedding space via Task A. This phase encourages the transformer to capture temporal dependencies in the linearly evolving latent space, without modifying the Koopman encoder. By doing so, we ensure that the latent representations remain fixed and in-

dependent from the sequence model. Training uses 64-step non-overlapping sequences, while validation uses longer 256-step sequences to assess generalisation across extended time horizons. The transformer employs a pre-LayerNorm architecture, with four attention heads [22, 23]. Teacher-forcing is applied during training using ground-truth embeddings for next-step prediction [24]. Loss is computed in the embedding space via MSE, as defined in Equation 7.

The final stage fine-tunes the frozen transformer for Task B: safety function prediction. Here, the transformer serves as a fixed backbone encoding the complete history of system dynamics. The prediction head, a multi-layer perception, receives as input the concatenation of the transformer's final hidden state and a 3D query state, producing a scalar safety score. This score quantifies the control effort required to keep the system within a predefined safety region. Ground-truth values are obtained via a recursive numerical method, using the same training sequences and data as Stage 2 but augmented with the corresponding safety values. Training optimises the MSE between predicted and ground-truth scores. Freezing the transformer ensures that safety head performance reflects the representational quality of

the pre-trained dynamics model. This design isolates the effect of representation transfer by assessing whether the pre-trained dynamics can generalise to a conceptually different prediction task.

## 3.4  Baseline Methods

To isolate the contributions of Koopman embeddings to downstream performance, we compare against three baseline configurations. Each model adheres to an identical training procedure, including data handling, sequence sampling, and transformer architecture, as detailed in Section 3.3. Moreover, a detailed overview of architectural components can be viewed in Appendix A.2. These baselines are referenced throughout the remainder of the paper as follows: Koopman (U) for the unfrozen transformer variant, PCA (PI) for the physics-informed PCA baseline, and PCA for the standard PCA baseline. Our main model of interest, the frozen Koopman-based transformer, is denoted as Koopman (F). We note that the Koopman (U) configuration mirrors the entire training process of Koopman (F), with the key distinction that the transformer's weights remain trainable during Task B. This setup enables the transformer to adapt its internal representation while the safety head learns to estimate control effort, thus assessing the benefit of end-to-end fine-tuning [7]. In the PCA (PI) baseline, the Koopman encoder-decoder is replaced by a physics-informed, multi-stage PCA pipeline.

Initially, a three-dimensional PCA is fit on the raw state vectors from the training set. This transformation is then applied not only to the system states but also to a collection of derived quantities, including first and second-order derivatives of the Lorenz system. These are mapped into a common coordinate frame and enriched with engineered features such as radial distance and angular velocity, yielding a nine-dimensional intermediate representation. A second PCA, trained on this extended dataset, reduces the features to match the embedding dimension used in the transformer. The resulting embedding captures domain-specific structure through physically meaningful components. While this approach benefits from strong prior knowledge, it lacks the generality and learning capacity of a trained encoder. The transformer and safety head are trained on this embedding using MSE in the predicted safety score. A complete overview of the feature engineering strategy is provided in Appendix A.3. Nevertheless, as noted by Shinn [25], PCA applied to smooth dynamical systems like the Lorenz attractor can produce spurious oscillatory patterns that may not reflect true latent structure.

Lastly, the standard PCA baseline serves as the most minimal configuration. Here, the system states are projected into a three-dimensional space that retains 100% of the original variance, with no physics-based augmentation [26]. While this embedding can assist transformer convergence by partially linearising the input space, the model struggled to converge when trained solely on the next-state prediction task (Task A). Despite this limitation, the baseline is retained to isolate the contribution of the training protocol itself-specifically, to assess how much of the downstream performance stems from the safety head alone. Like PCA (PI), this baseline is optimised using standard MSE loss between predicted and ground-truth safety values.

## 3.5  Evaluation Metrics

To quantify model performance across these stages, we employ three complementary metrics. MSE (as seen in Equation 7) is used as a primary loss and evaluation metric. Due to its squaring of error terms, MSE penalises larger deviations, which is particularly important in safety-critical scenarios where large prediction errors may indicate hazardous failures. Mean Absolute Error (MAE, i.e. Equation 8) is also reported, offering a more interpretable measure of average prediction deviation that is less sensitive to outliers. Finally, we compute the coefficient of determination ($R^2$, i.e. Equation 9), which captures the proportion of variance in the ground-truth safety values that is explained by the model's predictions. This metric provides a global view of how well the model reconstructs the overall safety landscape from Koopman-derived features.

$$\text{MAE}(y, \hat{y}) = \frac{1}{n} \sum_{i=1}^{n} |y_i - \hat{y}_i|, \tag{8}$$

$$R^2(y, \hat{y}) = 1 - \frac{\sum_{i=1}^{n} ||y_i - \hat{y}_i||^2}{\sum_{i=1}^{n} ||y_i - \bar{y}||^2}, \tag{9}$$

where $\bar{y}$ is the mean of all ground-truth values.

## 4  Results

Performance evaluation across 252 test trajectories reveals transfer learning advantages for Koopman embeddings over PCA-based baselines. We present a complete quantitative analysis across our chosen metrics, followed by spatial error analysis to identify possible patterns in model divergence across different regions of the Lorenz strange attractor that region $Q$ encompasses. Table 1 and 2 summarise quantitative metrics while Figure 4 maps error distributions for our qualitative analysis.

### 4.1  Quantitative Analysis

Our initial evaluation on the Task A pre-training revealed that the standard PCA model failed to converge on the forecasting task. We nevertheless

**Table 1.** Safety function prediction performance across models mean($\mu$) $\pm$ standard deviations($\sigma$). All metrics were computed on 252 test trajectories within the safety region. Statistical significance assessed via Wilcoxon signed-rank tests [27] with Bonferroni correction applied: $\alpha = 0.0083$ [28].

| Model | MSE ($\times 10^{-4}$) | MAE ($\times 10^{-2}$) | $R^2$ |
|---|---|---|---|
| **Koopman (F)** | **3.08 $\pm$ 6.55** | **1.16 $\pm$ 0.53** | **0.991 $\pm$ 0.089** |
| Koopman (U) | 5.59 $\pm$ 17.14 | 1.20 $\pm$ 0.59 | 0.989 $\pm$ 0.089 |
| PCA (PI) | 5.28 $\pm$ 8.83 | 1.48 $\pm$ 0.65 | 0.989 $\pm$ 0.090 |
| PCA | 16.80 $\pm$ 17.93 | 2.83 $\pm$ 0.98 | 0.983 $\pm$ 0.091 |

**Table 2.** Pairwise significance comparison ($p$-values) between models for control performance, assessed via Wilcoxon signed-rank tests with Bonferroni correction ($\alpha = 0.0083$). The results yielded identical significance outcomes for each pairwise comparison across all three evaluation metrics: MSE, MAE, and $R^2$. Below in the table, "ns" stands for not significant ($p > 0.05$).

| Model | Koopman (F) | Koopman (U) | PCA (PI) | PCA |
|---|---|---|---|---|
| Koopman (U) | **0.335 (ns)** | — | — | — |
| PCA (PI) | $<0.001$ | $<0.001$ | — | — |
| PCA | $<0.001$ | $<0.001$ | $<0.001$ | — |

retained this model for the final safety evaluation (Task B) to serve as a baseline, isolating the contribution of the neural network safety head. The full pre-training performance for all models is detailed in Appendix B.1.

Koopman (F), demonstrates superior performance across all metrics, achieving the lowest mean MSE of $(3.08 \pm 6.55) \times 10^{-4}$, the lowest MAE of $(1.16 \pm 0.53) \times 10^{-2}$, and the highest $R^2$ of $0.991 \pm 0.089$. The unfrozen Koopman variant, Koopman (U)'s accuracy comparably, with an MSE of $(5.59 \pm 17.14) \times 10^{-4}$, an MAE of $(1.20 \pm 0.59) \times 10^{-2}$, and an $R^2$ of $0.989 \pm 0.089$. However, Koopman (U) demonstrated less consistent performance, as seen with its standard deviation of 17.14. Both Koopman approaches significantly outperform the PCA-based baselines. PCA (PI) achieves intermediate performance with an MSE of $(5.28 \pm 8.83) \times 10^{-4}$, an MAE of $(1.48 \pm 0.65) \times 10^{-2}$, and an $R^2$ of $0.989 \pm 0.090$. Lastly, the baseline PCA shows the poorest capabilities, with an MSE of $(16.80 \pm 17.93) \times 10^{-4}$, an MAE of $(2.83 \pm 0.98) \times 10^{-2}$, and an $R^2$ of $0.983 \pm 0.091$. Statistical testing establishes a clear performance hierarchy in Table 2. Koopman (F) and Koopman (U) models demonstrate a statistically significant advantage over both the PCA (PI) and PCA baselines (all $p < 0.001$). Furthermore, the PCA (PI) model significantly outperforms the PCA model ($p < 0.001$). Notably, no significant difference exists between frozen and unfrozen Koopman approaches ($p = 0.335$), suggesting that representation quality drives transfer performance. We provide the complete pairwise statistical analysis in Appendix B.2, which covers the full twenty-one comparisons.

## 4.2 Qualitative Analysis

An important consideration when comparing the trained Koopman models to the PCA baselines is the potential of artefacts inherent to PCA when applied to time-series data [25]. These artefacts act as by-products rather than true features of the underlying system. For this reason, our PCA-based baseline may be susceptible to this effect. In particular, the PCA (PI) model, which includes time derivatives that may directly result in "phantom oscillations" that are not descriptive of the underlying dynamics of the system, but are nevertheless correlated [25]. Therefore, PCA (PI)'s performance may derive from fitting these PCA artefacts.

Figure 4 reveals the underlying mechanisms behind the performance differences observed in Table 1. The spatial distribution of prediction errors across the strange attractor provides insight into how different embedding methods capture the chaotic dynamics. Numbered velocity vectors (1-4, clockwise from top left) indicate the mean trajectory direction and magnitude at key regions. Across all models, we observe a notable concentration of error in the central transition region, where trajectories cross between the attractor's lobes (near x = 0). One likely cause is the high dynamical sensitivity in this area, where future states are most uncertain [13]. However, another possible explanation is related to the methodological design, as models were trained to evaluate strictly within region $Q$ - this transition boundary may result in unstable training. Visually, there is no significant difference in the error distribution between the Koopman (F) and Koopman (U) models, as indicated by the marginal histograms in Figure 4 that show the total error summed at each coordinate. PCA methods show greater error concentration in dynamically complex areas, such as regions of high curvature where trajectories revolve around the strange attractor (e.g. near vectors 1, 2, and 4). The interaction with the strange attractor [15] likely necessitates high sensitivity to conditions, resulting in an increased concentration of errors. The Lorenz system demonstrates dissipativity [13], meaning its trajectories, while chaotic, are bounded and converge to a finite strange attractor. This boundedness confines trajectories within a specific volume, which leads to a build-up of trajectory density around the attractor, creating regions of concentrated trajectories as seen by the consistent error space in Figure 4. These concentrated regions, combined with the inherent sensitivity to initial conditions within the strange attractor, may explain the observed error concentrations. This dissaptivity property is challenging for NNs to learn, as has also been highlighted by Tang et al. [30]. Furthermore, a stark contrast is evident between the two PCA-based methods. While the successfully pre-trained PCA (PI) model exhibits elevated errors, the base PCA

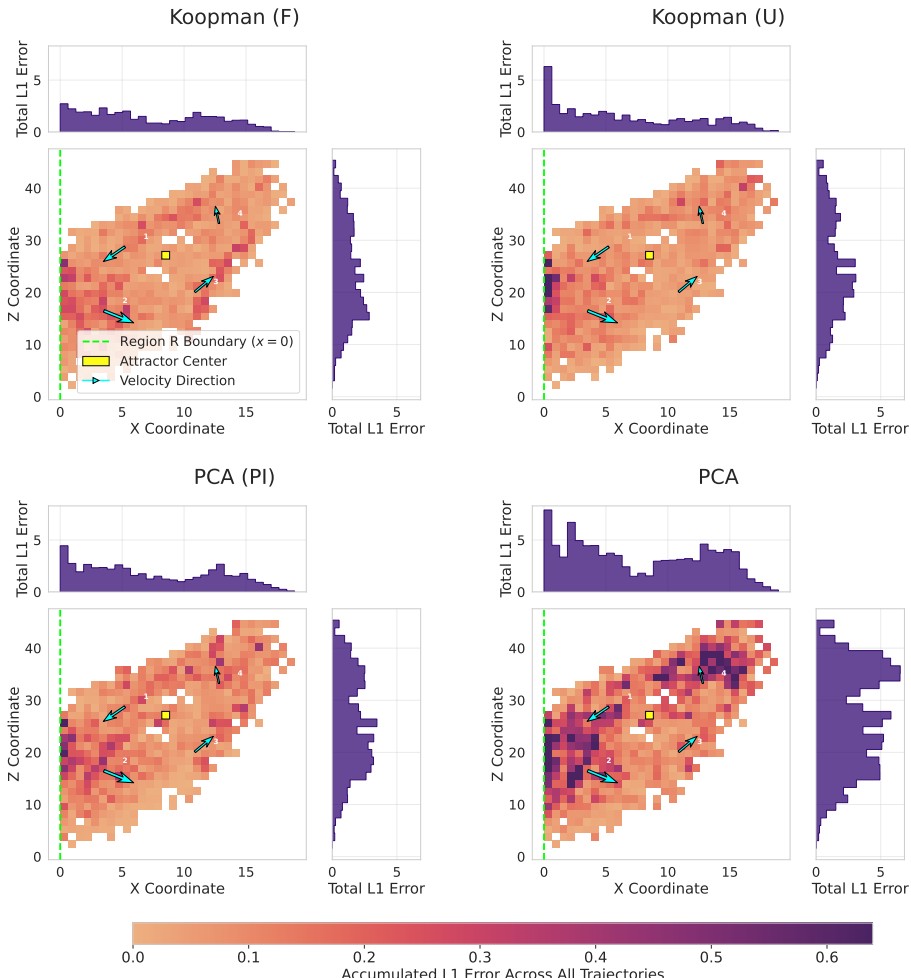

**Figure 4.** Comparative accumulated L1 error across the test dataset. The L1 error is defined as $L_1 = |y_{\text{true}} - y_{\text{pred}}|$. The figure displays the X–Z projection for four models. Light regions indicate low error; darker regions represent higher error concentrations. Contextual markers denote the boundary of the training region $Q$ ($s = 0$), corresponding to the Lorenz attractor's equilibrium point [29]. Vector lines show mean velocity and direction in each quadrant (see Appendix B.3 for details).

model displays significant errors with widespread issues, particularly across every vector region. As seen in Table 1, PCA achieved the worst results.

## 5   Conclusion

This work explored whether Koopman embeddings offer transferable representations of chaotic dynamics for safety-critical tasks. Our findings support this hypothesis: Koopman-based models consistently outperform Principal Component Analysis (PCA) baselines in downstream safety prediction, particularly in regions of high dynamical sensitivity. As shown in our quantitative and error analyses, the Koopman (F) and (U) models achieve higher control-safety performance and fewer high-risk errors. This advantage is likely due to their ability to encode generalisable dynamical structure, preserving key physical relationships across tasks [3]. Notably, this performance holds even when the transformer

is frozen, highlighting that Koopman embeddings capture genuine physical features rather than task-specific patterns, a hallmark of effective transfer learning [7, 31]. In practical terms, this allows for smaller, more efficient models. For instance, our Koopman autoencoder enables the use of a 4-layer transformer instead of an 11-layer PCA counterpart, reducing peak GPU usage by more than half during fine-tuning (see Appendix B.4). Finally, by leveraging pre-training and fine-tuning strategies inspired by Natural Language Processing, we demonstrated that transformer architectures can effectively model chaotic temporal dependencies such as those in the Lorenz system. Future work could apply these representations to formal safety methods like Control Barrier Functions [32], building upon recent theoretical advances in generalised Koopman operators for controlled systems to offer provable guarantees [5].

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

# A  Methodology Details

## A.1  Lorenz System

The following system of differential equations defines the Lorenz system:

$$\frac{dx}{dt} = \sigma(y - x); \tag{10}$$

$$\frac{dy}{dt} = x(\rho - z) - y; \tag{11}$$

$$\frac{dz}{dt} = xy - \beta z \tag{12}$$

Where $\sigma = 10$, $\rho = 28$, and $\beta = 8/3$ are the standard parameters settings that guarantee constant chaotic behaviour [13]. Trajectories from the Lorenz system are simulated using `scipy` [33], employing the RK45 integration method [34] with a time step of $dt = 0.01$. To ensure the reproducibility of chaotic dynamics, integration tolerances are set to `rtol` = $10^{-10}$ and `atol` = $10^{-10}$, respectively controlling the relative and absolute error of the solver [13].

## A.2  Model Architecture Specifications

The following architectures refer to our main Koopman (F) model. However, architectural structure is identical across all other baseline models with certain parameters differing, which are detailed throughout this section.

**The Koopman autoencoder:** Uses a symmetric architecture where encoder Input(3D) → Linear(3,500) → ReLU → Linear(500,32) → LayerNorm and decoder Linear(32,500) → ReLU → Linear(500,3) [23, 35].

**Transformer Implementation:** The transformer model implements a pre-LayerNorm decoder-only architecture. All models use GELU activation, and sinusoidal positional encoding [22, 36]. Weights are initialised from normal distribution $N(0, 0.05^2)$.

While the number of attention heads and layers differs across model types (detailed in Appendix A.6), all transformers follow the same general structure: input and output dimensions equal the embedding dimension, with feedforward layers scaled to $4 \times$ embedding_dim. The architecture processes embeddings through multiple transformer decoder layers, applies final layer normalisation, and uses a linear output projection that preserves the embedding dimensionality. Therefore a single transformer block follows: LayerNorm → MultiHeadAttention → Residual → LayerNorm → Linear(embedding_dim, 4×embedding_dim) → GELU → Linear(4×embedding_dim, embedding_dim) → Dropout → Residual. Where embedding_dim corresponds to the embedding dimension of the transformer's respective embedder.

Hence, the full transformer model follows the following structure: Input(embedding_dim) → PositionalEncoding → [TransformerBlock × N] → LayerNorm → Linear(embedding_dim, embedding_dim) → Output(embedding_dim). Where N refers to the transformer layers.

**Safety Function Implementation:** The safety predictor head processes the concatenated transformer final hidden state (32D) and query state (3D) through Linear(35, 128) → ReLU → Linear(128, 64) → ReLU → Linear(64,1).

## A.3  Physics-Informed PCA Feature Engineering

The features for the PCA (PI) baseline, shown in Table A.1, were constructed in three stages to create a 9-dimensional intermediate representation:

1. PCA Coordinates: The components $z_1, z_2, z_3$ represent the coordinates of the system in a new, decorrelated basis. They are obtained by applying a Principal Component Analysis (PCA) transformation to the original state vector $s$.

2. Transformed Derivatives: The time derivatives of the system's state, denoted $\dot{x}, \dot{y}$, and $\dot{z}$, are calculated using the Lorenz differential equations [13]. While the table shows this calculation in the original state space, the resulting velocity vectors are subsequently projected into the PCA basis to ensure they are represented consistently with the coordinates from step 1.

3. Engineered Features:

   a. The sine and cosine of the phase angle. This angle is calculated using the two-argument arctangent function, atan2($z_2, z_1$), which captures the angular position of the trajectory in the primary PCA plane.

   b. The radial distance from the origin in the principal plane, calculated as the Euclidean norm $\sqrt{z_1^2 + z_2^2}$. This feature captures the magnitude of the state's projection onto this plane.

## A.4  Safety Function Computation

The safety function $U_\infty(q)$ represents the minimum control effort required to maintain trajectories starting from state $q$ within a specified safe region $Q$

**Table A.1.** Component calculation for features in the Physics-Informed PCA baseline.

| # | Feature Group | Feature Components |
|---|---|---|
| 1. | **PCA Coordinates** | $z_1, z_2, z_3$ |
| 2. | **Transformed Derivatives** | $\dot{x} = \sigma(y - x)$ |
| | | $\dot{y} = x(\rho - z) - y$ |
| | | $\dot{z} = xy - \beta z$ |
| 3. | **Engineered Features** | $a)$   $\sin(\text{atan2}(z_2, z_1)),$ |
| | | $\cos(\text{atan2}(z_2, z_1))$ |
| | | $b)$   $\sqrt{z_1^2 + z_2^2}$ |

indefinitely [11]. The goal is to keep the system's trajectory within this region, regardless of whether it is affected by bounded disturbances.

The safety function is generated using a recursive numerical method known as a sculpting algorithm [10, 11], which is equivalent to value iteration for learning the global safe set. We follow the implementation from Valle et al. [12], who likewise applied this approach to a Lorenz system. The safety function is represented as a large matrix of safety values corresponding to a discrete state space. Each safety value represents the danger that a trajectory at that state poses in leaving the designated region.

### A.4.1   Discretisation

Due to computational limits, we discretise into a $30 \times 30 \times 30$ grid (27,000 points) spanning the safety region bounds, $x \in [0, 50]$, $y \in [-50, 50]$, $z \in [-50, 50]$. We calculate each state's safety value from its closest grid point and use linear interpolation to approximate the safety value at the state position. Furthermore, only disturbances within the safety region are considered valid. If a disturbed state lands outside the area, the norm is set to the distance to the closest point back to the grid, as is also done by Valle et al. [12].

### A.4.2   Disturbance Model ($\xi$)

The safety function is computed under a worst-case disturbance. We define the disturbance vector, $\xi \in \mathbb{R}^3$ as an additive noise term applied to the system dynamics $\dot{x} = f(x) + \xi$. The disturbance is generated from a hypercube, bounded by $[-0.1, 0.1]^3$. We sample 50 noise vectors uniformly at random from within this hypercube.

1. Apply all 50 sampled disturbances to compute next states, through Euler integration: $x' = x + (f(x) + \xi_i)\Delta t$, where $\Delta t = 0.01$.

2. For each disturbed state $x'$, we find the minimum cost (L2 norm) for the original state $C_i(x)$,

to reach any grid point $z$:

$$C_i(x) = \min_{z \in \text{Grid}} \max(\|x' - z\|_2, U_{prev}(z)), \quad (13)$$

where $U_{prev}(z)$ is its safety value from the previous iteration.

3. Assign the worst-case cost across all noise scenarios as a new safety value: $U_{new}(x) = \max_i C_i(x)$.

The iteration continues until $\max_x |U_{new}(x) - U_{prev}(x)| < 10^{-4}$ or 500 iterations are reached. In our case, convergence occurred after 36 iterations. Convergence is guaranteed as described by Sabuco et al. [10].

## A.5   Hyperparameter Optimisation

We ensure experimental fairness by comparing all models at their strongest configuration, evaluated at their best performance on the validation set. This implies stopping training once we've hit a considerable plateau in validation loss, yielding no further gain from optimisation or manual intervention.

For all new components and baselines without established precedents, we made use of the `Optuna` framework [37], which implements Bayesian optimisation with a Tree-structured Parzen Estimator (TPE) sampler [38].

Optimisation in Optuna makes use of an objective function, which returns a metric that the TPE sampler attempts to minimise. In addition, the objective function defines which hyperparameters will be optimised, the ranges of search and the steps of hyperparameter changes. For this reason, all objective functions started with wide ranges and were narrowed based on performance. On trials where the expected objective function return is lower than the best observed value, Optuna uses a median pruner that may interrupt a trial that under-performs compared to previous trials. Such trials are considered pruned.

### A.5.1 Task A: Next State Prediction

For Koopman Models, (F) & (U), we aimed to maintain consistency with prior work. For this reason, we used established hyperparameters from Geneva and Zabaras [4]. As our work serves as an extension of their methodology, retaining these settings mitigates the risk of introducing bias.

Conversely, the PCA and PCA (PI) baselines lacked established hyperparameter precedents for the next state prediction task. The objective function for all models was set to minimise validation loss.

Two main challenges influenced our optimisation process. Firstly, the search space was not uniform across all models, given that the backbone representations range from raw state inputs in PCA ($d = 3$) and physics-informed features in PCA (PI) ($d = 9$) to the high-dimensional Koopman embeddings ($d = 32$). Additionally, PCA baselines specifically proved to be significantly sensitive to hyperparameter selection, resulting in substantial time spent tuning. Frequently, Optuna's TPE sampler converged to regions of the search space with suboptimal performance, proposing similar configurations with minimal improvement, causing the median pruner to terminate trials early. To address all these issues, we performed exploratory runs to identify optimal hyperparameters that the automated search had difficulty identifying.

### A.5.2 Task B: Safety Head optimisation

For the downstream safety value prediction, all models used a similar Optuna procedure to what was used for PCA baselines in Task A. The objective function was based on the validation loss, consistent with Task A. Despite all safety heads following a similar architecture (a feed-forward neural network head), we also experienced varying degrees of convergence stability across models (failure to minimise loss). This is likely due to the varying sizes of the final hidden layers of the Task A pre-trained models. For this reason, the search space parameters for Optuna search still required manual tweaking.

## A.6 Training Hyperparameters

Hyperparameter configuration influences model learning and convergence for each specific task. More info on hyperparameter tuning can be seen in Appendix section A.5.

The learning rate (lr), which dictates the step size for weight updates, was varied to suit the specific requirements of each task. A learning-rate of $1 \times 10^{-3}$ was used for the Koopman autoencoder, while transformer pre-training used $1 \times 10^{-4}$ (Task A, Table A.2). Higher rates like $6.83 \times 10^{-3}$ and $1.04 \times 10^{-3}$ were found to be optimal for fine-tuning

the safety head in Task B (Table A.3). The training duration, defined by the number of epochs, was also tailored to each stage, ranging from 300 epochs for Task A training (Table A.2) to shorter periods, such as 90 epochs, for Task B (Table A.3). The batch size, or the number of samples per gradient update, was adjusted to balance gradient stability and memory constraints, with larger sizes, such as 512 (Table A.2), used for the autoencoder and a smaller size 32 (Table A.2, A.4), for the transformers. The Adam optimiser proved to be sufficient for all Task A training stages, with AdamW used for Task B training for Koopman (U) and PCA (PI) training [39, 40]. Architectural parameters define the model's capacity, including the embedding dimension, which sets the size of the latent space. This dimension was chosen to match the expected representational needs of each method: 32 for the Koopman models (Table A.2) to enforce a dense, high-dimensional representation where complex dynamics can be linearised; 9 for the PCA (PI) model (Table A.4), corresponding to the number of components created via its specific feature engineering; and 3 for the standard PCA baseline (Table A.5), matching the system's three physical dimensions. For transformers, the context length parameter was kept aligned with Geneva and Zabaras [4]'s implementation across all models with a context length of 64 (e.g. Table A.2). However, the number of transformer layers and attention heads were embedding type dependent. The final safety head's structure was defined by its layer dimensions. The Koopman autoencoder's training loss function used the following three loss weights [4]: $\lambda_0$ for state reconstruction, $\lambda_1$ for linear dynamics enforcement, and $\lambda_2$ for regularising the Koopman operator itself (Table A.2), supplemented by a minor weight decay to prevent overfitting.

Beyond the parameters optimised by Optuna, several configurations were fixed to ensure consistency. The transformer architecture implements a feed-forward network (FFN) dimension four times the size of its embedding dimension. For the Koopman autoencoder specifically, a distinct weight initialisation strategy was employed: while the encoder and decoder used standard Kaiming uniform initialisation, the Koopman matrix was deterministically initialised with linearly decreasing diagonal values and weakly coupled off-diagonals to promote stability.

Sequence generation also followed a set protocol: For Stage 1 (Koopman autoencoder), training used overlapping sequences of 64 steps with a 16-step stride. The safety head fine-tuning in Stage 3 used a similar protocol with the same 16-step stride. In contrast, Stage 2 (transformer pre-training) used non-overlapping 256-step sequences. For all models, validation and test sequences were longer (1,024 sequence steps) and non-overlapping to provide a

more challenging test of generalisation.

## A.7  Training Infrastructure

The implementation relies on `PyTorch` [41] with automatic mixed precision (AMP). A single NVIDIA 4070 (6GB) GPU is used for training, with the aid of `MLflow` [42] for experiment tracking, metric logging, and energy consumption tracking. To offset any possible over-optimisation, the test set was held until final evaluation.

**Table A.2.** Hyperparameters for Koopman Autoencoder Training and Transformer Pre-training (Task A).

| Parameter | Koopman Autoencoder | Task A |
|---|---|---|
| Learning rate (lr) | $1 \times 10^{-3}$ | $1 \times 10^{-4}$ |
| Number of epochs | 300 | 200 |
| Batch size | 512 | 16 |
| Optimiser type | Adam | Adam |
| Embedding dimension | 32 | 32 |
| Context length | - | 64 |
| Learning rate decay | 0.95 | - |
| Weight decay | $1 \times 10^{-8}$ | $1 \times 10^{-10}$ |
| Reconstruction Loss $\lambda_0$ | $1 \times 10^{4}$ | - |
| Dynamics Loss $\lambda_1$ | 1.0 | - |
| Regularisation Loss $\lambda_2$ | 0.1 | - |

**Table A.3.** Hyperparameters for Safety Head Fine-tuning (Task B) for Koopman (F) and Koopman (U) models.

| Parameter | Koopman (F) Task B | Koopman (U) Task B |
|---|---|---|
| Learning rate (lr) | $6.83 \times 10^{-3}$ | $1.04 \times 10^{-3}$ |
| Number of epochs | 80 | 50 |
| Batch size | 16 | 16 |
| Optimiser type | Adam | Adamw |
| Embedding dimension | 32 | 32 |
| Transformer layers | 4 | 4 |
| Transformer heads | 4 | 4 |
| Safety head layer 1 | 128 | 112 |
| Safety head layer 2 | 64 | 64 |

**Table A.4.** Hyperparameters for PCA (PI) Transformer Pre-training and Safety Head Fine-tuning.

| Parameter | Task A | Task B |
|---|---|---|
| Learning rate (lr) | $2.15 \times 10^{-3}$ | $7.52 \times 10^{-3}$ |
| Number of epochs | 300 | 90 |
| Batch size | 16 | 512 |
| Optimiser type | Adam | AdamW |
| Embedding dimension | 9 | 9 |
| Weight decay | $1 \times 10^{-10}$ | $1 \times 10^{-10}$ |
| Context length | 64 | - |
| Transformer layers | 11 | - |
| Transformer heads | 9 | - |
| Safety head layer 1 | - | 112 |
| Safety head layer 2 | - | 64 |

**Table A.5.** Hyperparameters for Standard PCA Baseline Pre-training and Safety Head Fine-tuning.

| Parameter | Task A | Task B |
|---|---|---|
| Learning rate (lr) | $1 \times 10^{-3}$ | $6.89 \times 10^{-3}$ |
| Number of epochs | 5 | 90 |
| Batch size | 16 | 512 |
| Optimiser type | Adam | Adam |
| Embedding dimension | 3 | 3 |
| Context length | 64 | - |
| Transformer layers | 3 | - |
| Transformer heads | 3 | - |
| Safety head layer 1 | - | 32 |
| Safety head layer 2 | - | 32 |

# B    Results Details

## B.1    Task A: Pre-training Performance

The performance of Task A, detailed in Table B.1, reveals differences in the capabilities of each embedding type. Our evaluation uses a continuous, 256-step auto-regressive rollout starting with just a single ground truth state. The MSE values of the reconstructed state are calculated over four consecutive 64-step windows [4].

The results indicate that the PCA model failed to converge to a useful predictive state. Its initial error of 121.35 is excessively high for a physical system and remains poor across the entire prediction horizon, indicating it was unable to learn the underlying dynamics. More revealing is the comparison between the two successful models. While the PCA (PI) model achieved the lowest initial error (62.33), its performance quickly degraded. The error nearly doubled in the subsequent window. This rapid "fall-off" suggests that while it learned short-term patterns, the representation was not stable for long-horizon forecasting.

In contrast, the Koopman model demonstrates long-term stability. Its prediction error remained consistently low and stable across all four horizons, never significantly exceeding an MSE score of 75.

## B.2    Complete Statistical Analysis

This section provides the complete pairwise statistical comparison results across all metrics for full transparency and reproducibility.

Table B.2 presents the statistical analysis underlying the summary results in Table 2. Our pairwise testing reveals a clear hierarchy. The Koopman (F) model's performance was compared to each of the other models. No statistically significant difference was found when comparing the Koopman (U) model to the Koopman model (F) across MSE ($p = 0.3355$), MAE ($p = 0.6897$), or $R^2$ ($p = 0.2625$). However, the Koopman (F) model significantly outperformed the PCA (PI) model on all three metrics: MSE ($p < 0.001$), MAE ($p < 0.001$), and $R^2$ ($p < 0.001$). Likewise, it demonstrated a significant advantage over the PCA model across MSE ($p < 0.001$), MAE ($p < 0.001$), and $R^2$ ($p < 0.001$).

Similarly, the Koopman (U) outperformed the PCA (PI) model across MSE ($p < 0.001$), MAE ($p < 0.001$), and $R^2$ ($p < 0.001$). It also performed significantly better than the PCA model on all metrics: MSE ($p < 0.001$), MAE ($p < 0.001$), and $R^2$ ($p < 0.001$).

Finally, the comparison between the two PCA-based methods revealed that the PCA (PI) model was significantly better than the PCA model across all three metrics: MSE ($p < 0.001$), MAE ($p < 0.001$), and $R^2$ ($p < 0.001$).

In summary, both Koopman models (F) and (U) significantly outperformed all PCA-based approaches. Among the PCA variants, PCA (PI) showed greater performance compared to the base PCA model, but inferior to the Koopman approaches.

## B.3    Velocity Vector Calculation for Error Accumulation Plot

The velocity vectors shown in the error accumulation analysis (Figure 4) are included to contextualise the error patterns with respect to the underlying dynamics of the Lorenz attractor. Their calculation follows a systematic procedure based on the test dataset after final result collection. Firstly, the two-dimensional X-Z state space is partitioned into four quadrants using the spatial median of the data points. For each of these four regions, the mean position (centroid) is computed to determine the vector's location. As such, the mean velocity of all states within that same quadrant is calculated. The resulting arrows in the figure indicate the predominant direction of the flow within each quadrant and size corresponds to the velocity's magnitude.

## B.4    Computational Resources

This section details the computational resources consumed during the key training stages. The metrics, collected via MLflow [42], offer insight into the computational cost associated with each approach by measuring peak memory usage and power consumption. We believe this data complements our performance analysis by providing a practical measure of the efficiency of each approach.

Table B.3 outlines peak resource consumption across all training stages. The data reveals a clear trade-off between upfront training cost and downstream efficiency. During stage 1 training, the Koopman autoencoder required intensive computation with peak power consumption of 56.4 W. However, this initial investment pays off during fine-tuning: the Koopman (F) model consumed only 24.6 W peak power, while the PCA (PI) model hit a peak of 59.8 W - more than double the Koopman fine-tuning consumption.

Even though PCA-based models do not require any sort of training, we observe lower peak wattage during transformer training: Koopman stage 2 consumed 45.0W compared to PCA PI's 55.3W.

**Table B.1.** Transformer pre-training performance on Task A. Values represent the reconstructed State MSE, calculated over sequential 64-step windows of a single 256-step auto-regressive prediction. Convergence was determined based on the stability of error.

| Embedding Type | State MSE per Prediction Horizon (Steps) | | | | Converged? |
|---|---|---|---|---|---|
| | [0-64) | [64-128) | [128-192) | [192-256) | |
| Koopman | 73.75 | 75.34 | 74.65 | 73.86 | Yes |
| PCA (PI) | 62.33 | 121.15 | 116.66 | 116.41 | Yes |
| PCA | 121.35 | 133.48 | 140.90 | 137.06 | No |

**Table B.2.** Complete pairwise statistical comparison results across all metrics. P-values from Wilcoxon signed-rank tests with Bonferroni correction ($\alpha = 0.0083$).

| Method 1 | Method 2 | Metric | P-value | Significant | Winner |
|---|---|---|---|---|---|
| Koopman (F) | Koopman (U) | MSE | 0.3355 | No | — |
| Koopman (F) | Koopman (U) | MAE | 0.6897 | No | — |
| Koopman (F) | Koopman (U) | $R^2$ | 0.2625 | No | — |
| Koopman (F) | PCA (PI) | MSE | <0.001 | Yes | Koopman (F) |
| Koopman (F) | PCA (PI) | MAE | <0.001 | Yes | Koopman (F) |
| Koopman (F) | PCA (PI) | $R^2$ | <0.001 | Yes | Koopman (F) |
| Koopman (F) | PCA | MSE | <0.001 | Yes | Koopman (F) |
| Koopman (F) | PCA | MAE | <0.001 | Yes | Koopman (F) |
| Koopman (F) | PCA | $R^2$ | <0.001 | Yes | Koopman (F) |
| Koopman (U) | PCA (PI) | MSE | <0.001 | Yes | Koopman (U) |
| Koopman (U) | PCA (PI) | MAE | <0.001 | Yes | Koopman (U) |
| Koopman (U) | PCA (PI) | $R^2$ | <0.001 | Yes | Koopman (U) |
| Koopman (U) | PCA | MSE | <0.001 | Yes | Koopman (U) |
| Koopman (U) | PCA | MAE | <0.001 | Yes | Koopman (U) |
| Koopman (U) | PCA | $R^2$ | <0.001 | Yes | Koopman (U) |
| PCA (PI) | PCA | MSE | <0.001 | Yes | PCA (PI) |
| PCA (PI) | PCA | MAE | <0.001 | Yes | PCA (PI) |
| PCA (PI) | PCA | $R^2$ | <0.001 | Yes | PCA (PI) |

**Table B.3.** Peak computational resource usage during training.

| Training Stage / Model | GPU Power (W) | GPU RAM (MB) | System RAM (GB) |
| --- | --- | --- | --- |
| *Stage 1: Autoencoder Training* | | | |
| Koopman Autoencoder | 56.4 | 1919.2 | 23537.1 |
| *Stage 2: Transformer Pre-training (Task A)* | | | |
| Koopman Transformer | 45.0 | 1574.0 | 22827.7 |
| PCA (PI) Transformer | 55.3 | 756.8 | 7292.7 |
| PCA Transformer | 17.7 | 626.8 | 9114.6 |
| *Stage 3: Safety Head Fine-tuning (Task B)* | | | |
| Koopman (F) | 24.6 | 746.5 | 13826.3 |
| Koopman (U) | 28.4 | 740.8 | 13196.7 |
| PCA (PI) | 59.8 | 954.0 | 13560.8 |
| PCA | 59.4 | 882.0 | 13072.8 |

