# OpenReview forum: "On the Generalisation of Koopman Representations for Chaotic System Control"
_NLDL.org/2026/Conference — NLDL 2026 Spotlight_

### Official Review · Reviewer_MyzZ · 2025-10-08
**On the Generalisation of Koopman Representations for Chaotic System Control**

**Rating:** 4
**Confidence:** 3

**Summary:**

This paper presents a method to learn representations for chaotic systems. The method consists of learning an embedding to create a linear operator on the system. The general principle is to frame the problem as a next state prediction problem and use a transformer model for sequential next token prediction task. Their training pipeline has three steps. First, they train an auto encoder to learn a linear representation of the data. Second, they train the transformer model. Finally, they fine tune their model. The goal of their method is to both perform a next state prediction but also to learn a safety value of the energy required to keep the state in a safe area. They use prior methods to decompose the embedding with a structured strategy. They find that their method of using embeddings and a transformer backend is able to improve upon baselines of PCA in safety function prediction.

The overall soundness and correctness of this paper seems to be reasonable. I didn't see any issues with the work where I would question these aspects of it.

**Strengths:**

The correctness of this work seems reasonable. I didn't find anything wrong with the ideas presented in the paper.

This paper provides a relevant work for this domain. This method is an interesting framing of the problem and shows superior results.

The presentation of this work is good. This is a relatively complex topic but the authors do a good job of presenting their work. Their figures are clear and helpful and the writing is clear.

As far as I can tell, this work is correct and provides statistically significant results like CIs. The method is well motivated and clear.

**Weaknesses:**

There are a few spelling errors in the work so I would recommend checking it with a spell checker before camera ready or resubmission.

The significance is hard to judge. This is the weakest part of the work not because the originality is low but instead because they do not have a related works section to compare their method and strategy to other methods. I think this is a major problem, and would highly recommend adding this to show the novelty of the method and provide the reader with context within the broader scope of the literature.

**Justification:**

I believe that this paper focuses an good research question and provides an interesting method. Their method is well described and I was able to understand it well. Their results showed that their method is able to decompose the non linear dynamics into an easier to learn structure which is the goal of the work. I found the only major issue to be a lack of related work. If some information from the main paper has to be moved to the appendix to fit it that is ok, since it is critical in my opinion. I still put accept since I think the paper is well done but this does need to be added before camera ready

---

> ### Author Rebuttal · Authors · 2025-10-20
>
> **Reviewer's concern 1:** *There are a few spelling errors in the work so I would recommend checking it with a spell checker before camera ready or resubmission.*
>
> **Reply:**
> Thank you for this feedback. We will ensure to take another thorough look through the entire paper. Especially noting possible inconsistencies between authors' writing styles and GB/US English, which may have led to such mistakes.
>
> ---
>
> **Reviewer's concern 2:** *The significance is hard to judge. This is the weakest part of the work not because the originality is low but instead because they do not have a related works section to compare their method and strategy to other methods. I think this is a major problem, and would highly recommend adding this to show the novelty of the method and provide the reader with context within the broader scope of the literature.*
>
> **Reply:**
> We appreciate the comment regarding how to strengthen the work further, especially as it pertains to our structure.
>
> Throughout the paper we aimed to contextualise our work and highlight its novelty in our introduction and preliminaries section. This structural approach was necessary to present the relevant ideas in a clear and progressive manner. We found this necessary as research in Physics-informed models spreads over many fields, with plenty of work done within the domains of applied physics and mathematics. For this reason, we pushed to position our work by discussing the foundations of NNs for chaos modelling, and the current state-of-the-art for chaotic system modelling. Our methodology is a direct extension of the work by Geneva & Zabaras  (2022), further applying existing ML frameworks to a traditionally physics domain task.
>
> We understand how the omission of a relevant work section makes it harder at a glance. Regarding the generalisability of models across tasks, to our knowledge, no study has been done using a comparable model architecture and within a solely dynamical system framework. Hence, the core question of this paper addressing this research gap.
>
> **Reference:**
> N. Geneva and N. Zabaras. “Transformers for modeling physical systems”. In: Neural Networks 146 (2022), pp. 272–289.

---

### Official Review · Reviewer_xDwC · 2025-10-10
**Transfer Learning Results for Koopman Embeddings in Chaotic System Control**

**Rating:** 4
**Confidence:** 5

**Summary:**

This paper examines the transferability of Koopman-based representations learned via next-state prediction to safety-critical control tasks in chaotic dynamical systems. Using the Lorenz system as a benchmark, the authors propose a three-stage approach: (1) learning Koopman embeddings through autoencoding with structured operator constraints, (2) pre-training a transformer on next-state prediction (Task A), and (3) fine-tuning for safety function approximation (Task B). The key contribution is showing that Koopman embeddings enable effective transfer from prediction to control. Freezing pre-trained transformer weights during fine-tuning yields no performance drop compared to full fine-tuning, suggesting the representations capture reusable dynamical structure rather than task-specific features. Both frozen and unfrozen Koopman models outperform standard and physics-informed PCA baselines across all metrics (MSE, MAE, R²). The work draws a novel parallel to NLP transfer learning, where next-token prediction underlies downstream generalization, extending this idea to chaotic systems.

**Strengths:**

* Draws a novel parallel between NLP transfer learning (next-token prediction) and chaotic systems (next-state prediction), offering strong theoretical grounding.
* Frozen transformer weights perform on par with unfrozen ones (Koopman F vs. Koopman U, p = 0.335), indicating learned representations capture physical rather than task-specific structure.
* Structured Koopman operator encodes dissipation (diagonal) and rotational dynamics (skew-symmetric bands), providing strong inductive bias.
* Includes standard PCA and physics-informed PCA_PI (with 9 engineered features) for fair ablation and comparison.

**Weaknesses:**

* Results are limited to the Lorenz system; broader claims would be stronger with tests on additional chaotic systems (e.g., Rössler, Chen)
* Koopman models underwent Optuna-based optimization; it’s unclear if baselines received equal tuning or architectural search depth.
* “Phantom oscillations” in PCA (Sec. 4.2) are noted but not analyzed—unclear whether gaps stem from artifacts or inherent model limits.

**Justification:**

This paper makes a substantial contribution to transfer learning in chaotic dynamical systems. Its core finding—that frozen transformer weights perform on par with unfrozen ones—demonstrates representational transfer of physical structure. Koopman models significantly outperform PCA baselines with statistical validation and notable computational efficiency.  Despite being limited to the Lorenz system, the results are robust and show clear relevance to the ML and dynamical systems communities.

---

> ### Author Rebuttal · Authors · 2025-10-20
>
> **Reviewer's concern 1:** *Results are limited to the Lorenz system; broader claims would be stronger with tests on additional chaotic systems (e.g., Rössler, Chen)*
>
> **Reply:**
> Thank you for your valuable suggestion. We absolutely agree that applying this methodology to other chaotic systems (such as Rössler or Chen) is an important direction for future work.
>
> For this paper our focus was to use a single, fundamental benchmark to isolate and test the central hypothesis. This hypothesis revolves around within-domain task generalisation - whether representations learned for a prediction task can be reused for a complex downstream control task. We believe extending this to cross-domains would dilute the core contributions.
>
> We chose the Lorenz system because it's a well-known non-linear system, with control tasks available to serve as downstream tasks. Formulating comparable downstream control tasks for other systems would greatly expand the scope of the paper with a weaker quantitative comparison.
>
> For these reasons, we prioritised a focused investigation on a single system to test our specific hypothesis. Future extension to cross-domain systems or as mentioned in our conclusion to applied control problems such as Control Barrier Functions, are excellent follow-up research questions.
>
> We will address the comment in the camera-ready by making clearer emphasis in the task generalization framework in the abstract and introduction.
>
> ---
>
> **Reviewer's concern 2:** *Koopman models underwent Optuna-based optimization; it’s unclear if baselines received equal tuning or architectural search depth.*
>
> **Reply:**
> We thank the reviewer for raising this critical point regarding experimental fairness. We propose adding to the Appendix (A.5) to clarify the optimisation process further.
>
> Due to the availability of existing research, Transformers (F) and (U), used pre-existing hyperparameters for Task A, next state prediction. The PCA baselines however, did not have existing precedent and required manual tuning of the Optuna search parameters to find models that performed adequately on validation loss.
>
> To reiterate:
> 1. Task A: Koopman models adopted established hyperparameters set by Geneva & Zabaras, 2022, as our work is a direct extension of that methodology. For PCA, Optuna search with manual intervention was used to re-configure the search space to help the optimiser escape "prune feedback loops", where the models failed to learn.
> 2. Task B: To ensure a fair evaluation all models underwent separate Optuna search on their safety heads. This too required manual configuration to search parameters to account for backbone complexity.
>
> Our goal was to ensure all models were tuned until their validation loss clearly plateaued, with no further improvements stemming from Optuna without a pruning feed-back loop. The PCA and PCA-PI models have different input dimensions and architectures, leading to different search space characteristics that require altering search parameters. Of key differences these included the differing feature complexity across baseline models: as the PCA model used raw input states (3 input features) but had trouble converging; while the PCA PI used physics informed features (9 input features) which greatly added complexity; both Koopman (F) and (U) used a stable input of Koopman embeddings (32 input features).
>
> As mentioned in our Appendix (A.5) & (A.6), Optuna was used to explore the hyperparameter space to identify optimal search parameters. Certain parameters such weight decay, were left out of search as they lead to instability. To ensure over-fitting was to a minimum we utilised a held-out test set to compile our final results.
>
> ---
>
> **Reviewer's concern 3:** *“Phantom oscillations” in PCA (Sec. 4.2) are noted but not analyzed—unclear whether gaps stem from artifacts or inherent model limits.*
>
> **Reply:**
> Thank you for this point, it illustrates the basis of our baseline comparison. Our paper's position is that the performance gaps stem from the artefacts which themselves are inherent limits of the PCA-based models. These phantom oscillations are statistical by-products of the system's dynamics. While correlated with the dynamics are not true features. This is a well known issue when applying PCA to smooth time-series data. Therefore, this was in particular an issue for the PCA (PI) model which included time derivatives. While it started off with competitive results at a lower horizon [0-64) in Task A: 62.33 MSE, as seen in appendix B.1. It failed at longer time horizons (>64) dropping to around approximately 120 MSE. This lack of long term stability is why we hypothesised a lower accuracy for Task B results.
>
> While a further analysis to what extent these oscillations impact performance would be highly useful. We deem this to be future work evaluating the efficacy of PCA based models on time-series data more broadly.
>
> **Reference:**
> N. Geneva and N. Zabaras. “Transformers for modeling physical systems”. In: Neural Networks 146 (2022), pp. 272–289.

---

### Official Review · Reviewer_MsmV · 2025-10-12
**Interesting Take on ML methods for chaotic Lorenz system**

**Rating:** 4
**Confidence:** 2

**Summary:**

The paper suggests a method for studying chaotic systems, inspired by next-token prediction in NLP tasks, where here the task becomes to predict the next state of the system. The authors use Koopman embeddings/representations and study the Lorenz system via a 3-phase method:  they first learn Koopman embeddings through autoencoding, by pretraining a transformer for next-state prediction, and fine-tuning for safety-critical control. The main finding is that their method outperforms PCA and physics-informed PCA, highlighting their approach is effective for for multitask learning in various physics-ML problems.

At a technical level, the authors propose some variants of Koopman models, and proceed by comparing the frozen Koopman model with an unfrozen variant and with the two PCA baselines, the standard one (simple vanilla PCA) and a physics‑informed version of PCA (PCA‑PI). On the reported metrics, Koopman‑Frozen performs best and as they note, freezing the Transformer does not significantly degrade performance relative to fine‑tuning.

**Strengths:**

Overall, I like the paper's setting and I believe it gives a clear and principled setup to study the Koopman embeddings for a well-motivated physics problem.

I also find interesting the fact that as it seems, the frozen and unfrozen variatns are similar in performance: this means that benefits are come from the representation, not a task‑specific adaptation.

Also, extensive evaluations on their Lorenz task and comparisons with PCA variants are convincing.

**Weaknesses:**

My main concern with the paper, even though I like the work, is that it focuses overly on one exapmle about the Lorenz system as the main form of chaotic systems. But this is just one system and given that the paper discusses generalization on chaotic systems control I was expecting a broader study, perhaps with other types of chaotic systems as well.

I suggest that the authors add at least one more system in their repertoire to highlight truly the strength of their method.

**Justification:**

As I mentioned, the benefits of the paper overall are more convincing than the weakness, even though having more successful examples would strengthen the main message of the paper.

---

> ### Author Rebuttal · Authors · 2025-10-20
>
> **Reviewer's concern:** *My main concern with the paper, even though I like the work, is that it focuses overly on one example about the Lorenz system as the main form of chaotic systems. But this is just one system and given that the paper discusses generalization on chaotic systems control I was expecting a broader study, perhaps with other types of chaotic systems as well.
> I suggest that the authors add at least one more system in their repertoire to highlight truly the strength of their method.*
>
> **Reply:** Thank you for your comment. We agree that our study focuses on a single chaotic system.
>
> We intended for our paper to evaluate task generalisation foremost, not system generalisation, by picking a well-known dynamic system to isolate the specific research question we were addressing. We understand how this confusion may arise with the focus of generalisation. We aimed to made this scope clear in the abstract and introduction, where the main identified research gap is: "the extent to which the resulting representations support downstream tasks such as control remains an open question."
>
> We made this choice as previous work such as (Geneva & Zabaras, 2022), has already established that this type of Koopman-transformer architecture is effective for modelling different physical systems (Lorenz, Navier-Stokes, 3D reaction diffusion systems). Given the architecture's applicability across different systems of varying complexity but equally impressive results has already been demonstrated, our next question was to investigate its generalisability across tasks within a single domain.
>
> Without doubt we agree that applying this methodology to other chaotic systems is a valuable direction for future work. We will address the comment in the camera-ready by making clearer emphasis in the task generalization framework.
>
> **Reference:**
> N. Geneva and N. Zabaras. “Transformers for modeling physical systems”. In: Neural Networks 146 (2022), pp. 272–289.

---

### Official Review · Reviewer_FT8w · 2025-10-13
**The contribution is correct and interesting**

**Rating:** 5
**Confidence:** 3
**Final Rating:** 5
**Final Confidence:** 3

**Summary:**

In this work, the authors train an autoencoder to learn Koopman representations of the trajectories of the Lorenz system. They then use the states from the autoencoder's latent space to train a transformer for a specific downstream task: safety function prediction.

**Strengths:**

The paper is well-structured and written, making it easy to follow the reasoning.

Its contributions are clear from the beginning; they are nicely laid out within the relevant context, and they are not overemphasised, making it clear what value they add to the existing literature.

As far as I'm aware, the addition of Koopman in the context of safety function prediction is new.

The claims they make are well supported by experiments on the Lorenz system, with plenty of details on the framework they used, and where each choice is further supported by ablation studies.

**Weaknesses:**

I only have a few minor comments to make:
1. While describing the Koopman operator, one could mention the trade-off of this approach: you get a simpler linear description of the system, at the expense of a higher-dimensional state space. When choosing the dimension of the latent space, i.e. the dimension of the Koopman representation (32), have you noticed any particular patterns? Apart from hurting the computational cost, does increasing that dimension improve the performance?
2. The description of how to build the ground truth for the safety function is a bit hard. Maybe you could expand it further in the Appendix, by better explaining the algorithm of equation (3). For example, you could clarify the type of noise $\xi$ used, or why that algorithm converges.
3. To support the previous point, the description of Figure 1 could also be improved:
    - For example, you introduce some terms there, like *query points* or *hypothetical states*, which are not referred to in the text. It would be nice to refer to the exact symbols used in the formulas instead.
    - Also, I understand that the coloured dots represent the ground truth, calculated as per equation (3). But the safety function is computed on a grid of points, as you also mention in section 3.2, so why isn't it represented as a grid also in figure 1?
4. When you introduce the Koopman embedding in section 3.1, you talk about the autoencoder, but you only describe what it is in Appendix A.2. You could either describe briefly what it is composed of, or at least cite the appendix.
5. Have you done an ablation study for the particular structure of $\mathbf{K}$ of equation (5)? Is it really necessary?
6. I think also figure 3 could be more consistent:
    - In stage 1, the Koopman embeddings are described with $y_t$, while in stage 2 and 3 they are called $z_t$. My understanding is that they are the same object. If that's the case, you could use the same letters.
    - In stage 1 again, visually, the figure seems to suggest that the encoder is an MLP and the decoder the identity.
    - In stage 2, the input sequence is described with boxes, while in stage 1, you had just $s_t$.
7. In table 2, there is I think a typo in the first entry (ns?), and you should also specify which metric you used to compute those $p$-values.

**Final Justification:**

I only had minor, mostly aesthetic, concerns to begin with, which the authors promise to address for the camera-ready version.

I therefore confirm my initial rating.

**Justification:**

I find the paper well written and well supported by the analysis and the experiments.
I set a lower confidence vote, as I'm not 100% confident in the literature about safety function prediction.

---

> ### Author Rebuttal · Authors · 2025-10-20
>
> 1. *While describing the Koopman operator, one could mention the trade-off of this approach: you get a simpler linear description of the system, at the expense of a higher-dimensional state space. When choosing the dimension of the latent space, i.e. the dimension of the Koopman representation (32), have you noticed any particular patterns? Apart from hurting the computational cost, does increasing that dimension improve the performance?*
>
> **Reply:**
> The aforementioned trade-off is an excellent point, we agree that as the Koopman operator introduces a trade-off: As the goal is to achieve a simpler linear description there is an implicit trade-off with a higher dimensional space. We would like to add that the primary advantage of a simplified description of the system is a model that is able to capture physical structure (explaining greater variance), which in turn leads to stronger performance in downstream tasks such as safety value prediction. Increasing Koopman dimensionality would not necessarily create a "simpler" linear description that would add to increased performance.
>
> While we did not conduct a formal ablation study on the dimensionality of the operator in the current work, we acknowledge that such an analysis is a valuable direction for further investigation of its impact on performance. We can hypothesise that while increasing the dimensionality may increase performance for the intermediate Next State Prediction task ("Task A"), this added complexity may contribute to substantial over-fitting that would decrease downstream performance for the Safety Value Prediction task ("Task B").
>
> Our selection of a 32-dimensional space was guided by previous foundational work in this area (Geneva & Zabaras, 2022). Following this work, it has been seen that for more complex dynamical systems, such as "2D Fluid Dynamics" and "Gray-scott" systems, significantly greater dimensions were required to produce competitive models (128, 512 respectively). In general the original authors scaled dimensionality as input size increased.
>
> ---
>
> 2. *The description of how to build the ground truth for the safety function is a bit hard. Maybe you could expand it further in the Appendix, by better explaining the algorithm of equation (3). For example, you could clarify the type of noise $\xi$ used, or why that algorithm converges.*
>
> **Reply:**
> Thank you for the valuable feedback. We agree that certainly a more detailed explanation of the ground truth would improve clarity. The algorithm is a value iteration algorithm, which iteratively updates the safety values for each grid point by computing the "worst-case" outcome.
>
> The type of noise used $\xi$ is generated from a hypercube. In our implementation the disturbance vector is bounded by `[-0.1,0.1]`. We sample 50 noise vectors uniformly at random from within this hypercube selecting the one that results in the highest required safety cost (This safety cost is defined as the minimum control magnitude, an L2 norm, required to stay safe after that disturbance).
>
> Convergence is guaranteed as described by (Sabuco, Sanjuán, & Yorke, 2012), and can be understood as a "sculpting algorithm". As we iterate, the algorithm "removes" points (by assigning them a high cost) if, after a worst-case disturbance, there is no safe and reachable destination. The iteration stops when the safety values no longer change.
>
> ---
>
> 3. *To support the previous point, the description of Figure 1 could also be improved: a) For example, you introduce some terms there, like query points or hypothetical states, which are not referred to in the text. It would be nice to refer to the exact symbols used in the formulas instead. b) Also, I understand that the coloured dots represent the ground truth, calculated as per equation (3). But the safety function is computed on a grid of points, as you also mention in section 3.2, so why isn't it represented as a grid also in figure 1?*
>
> **Reply:**
> Thank you for the constructive feedback on Figure 1. We agree that its description and connection to the text can be improved. We aim to do it for the camera-ready version.
>
> Regarding a) Query point is synonymous to our usage of query state throughout the rest of the paper. We'll make changes to ensure the figure better aligns with the terminology present in the rest. We will also clarify that the "Hypothetical States" represent the set of possible future states $\{q_n\}$ that evolve from $q$.
>
> For b), Your understanding of (3) is valid and the safety function is indeed computed on a grid.  A completely literal representation would be a static heat-map of those final grid values with no trajectories plotted.  Our choice for a side-by-side scatter plot comparison was to illustrate the nature of the task more so than the algorithmic process of safety value creation. This being the contrast of the local nature of next-state prediction (Task A) to the long-horizon (global) nature of the safety function (Task B). Given that the main focus of our paper is on machine learning making the most of Koopman embeddings, we believe this conceptual representation is better aligned with the paper's objectives. We are confident that by standardising the terminology as described above, the figure will be clearer and more explicit.
>
> ---
> 4. *When you introduce the Koopman embedding in section 3.1, you talk about the autoencoder, but you only describe what it is in Appendix A.2. You could either describe briefly what it is composed of, or at least cite the appendix.*
>
> *Reply:*
> Thank you for this helpful suggestion. We agree adding an explicit reference to the appendix in Section 3.1 will improve the technical clarity of the paper. This will be added to our final version.
>
> Our initial brevity was based on the model following a standard auto-encoder design (illustrated in Stage 1 of Figure 3) that mirrors the dimension transformations presented in section 3.1, and its foundation in previous work this paper is based on (Geneva & Zabaras, 2022).
>
> ---
>
> 5. *Have you done an ablation study for the particular structure of $K$ of equation (5)? Is it really necessary?*
>
> **Reply:**
> Thank you for your interest in the structure of $K$. We did consider performing an ablation study as a comparative baseline, but we did not deem it necessary for two main reasons:
>
> 1) The $K = D+ S_{band}$ structure is foundational and theoretically motivated. As established in Budisic et al., (2012), this $K$ structure is a critical application in achieving an effective non-linearised matrix for ML systems. The goal of this structure is to reduce model complexity without any information loss. This is necessary to avoid over-fitting as it properly encodes a physically possible representation, which allows for skill transfer to downstream tasks that rely on infinite horizons. It's unlikely that a full dense matrix would be able to capture such dynamics stably.
> 2) Our existing baselines already serve as a good surrogate for this ablation. We can see that in our PCA-based baseline models, performance was severely degraded when using non-inductive bias. Both PCA and PCA (PI) showed unstable long-term forecasting and were significantly outperformed on the final control task. We believe an ablation of the $K$ would suffer a similar fate.
>
> ---
>
> 6. *I think also figure 3 could be more consistent: a) In stage 1, the Koopman embeddings are described with y_t, while in stage 2 and 3 they are called z_t. My understanding is that they are the same object. If that's the case, you could use the same letters. b) In stage 1 again, visually, the figure seems to suggest that the encoder is an MLP and the decoder the identity. c) In stage 2, the input sequence is described with boxes, while in stage 1, you had just s_t.*
>
> **Reply:**
> We appreciate the suggestions for figure consistency:
>
> a)  Your concerns are valid here; both represent the same object. The latent embedding is produced by the encoder $\phi_e$. Our different labels were intended to separate the "evolved state" variable in Stage 1 from the input sequence in Stage 2. We'll update the figure to use $y_t$ consistently for all embeddings.
>
> b) You are right that the visual for the decoder $\phi_d$ is misleading. The decoder is also an MLP with a hidden layer. We'll fix the diagram to include this.
>
> c) We understand how this can be confusing; however, the key difference between Stage 1 and Stage 2 is that the former operates on a single state at a time (to learn the one-step dynamics). While the latter is a Transformer model that takes an input vector of the full sequence. We felt that repeating the sequence notation would be visually redundant and clutter the diagram. We also clarify this in the caption when referring to the "sequence of input states".
>
> ---
>
> 7. *In table 2, there is I think a typo in the first entry (ns?), and you should also specify which metric you used to compute those p-values.*
>
> **Reply:**
> Thank you for the suggestions to improve the clarity of Table 2:
>
> 1) The inclusion of (ns) is not a typo. It was intended to stand for "non-significant"; we'll ensure this is clarified in the caption.
> 2) Regarding the metric used, we apologise if this was unclear. Per the caption, the significance results were "consistent for each pairwise comparison across all three evaluation metrics: MSE, MAE, and R^2". This means that the p-value comparing Koopman (F) to PCA (PI) used all three metrics, and was significant across all three: MSE (< 0.001), MAE (<0.001), and R^2(<0.001). We'll revise the caption to make this point clearer.
>
> **References:**
> * J. Sabuco, M. A. Sanju´an, and J. A. Yorke. “Dynamics of partial control”. In: Chaos: An Interdisciplinary Journal of Nonlinear Science (2012)
> * M. Budisic, R. Mohr, and I. Mezic. “Applied koopmanism”. In: Chaos: An Interdisciplinary Journal of Nonlinear Science (2012).
> * N. Geneva and N. Zabaras. “Transformers for modeling physical systems”. In: Neural Networks 146 (2022).

---

### Meta-Review · Area_Chair_u9UM · 2025-10-29

**Recommendation:** Accept (Poster)
**Confidence:** 4

**Metareview:**

The paper presents an analysis of representations based on Koopman theory for the prediction and control of chaotic dynamical systems.

The reviewers agree that the paper is interesting and technically sound. The novelty of the transferability analysis was also appreciated. Some concerns were raised regarding the limitations of the empirical analysis (only the Lorenz system is considered). Nevertheless, the reviewers agree that the paper represents a valuable contribution to the conference. There were also minor concerns about the presentation of certain aspects, but the rebuttal addressed them convincingly.

Overall, this is a good contribution, and I recommend its acceptance.

---

### Decision · Program_Chairs · 2025-11-05

**Decision:**

Accept (Spotlight)

**Comment:**

We recommend an oral and a poster presentation given the AC and reviewers recommendations.

A spotlight presentation refers to a poster selected for an oral highlight but not designated as a full oral presentation per the AC’s recommendation.